# Expression of Ceramide-Metabolizing Enzymes in the Heart Adipose Tissue of Cardiovascular Disease Patients

**DOI:** 10.3390/ijms24119494

**Published:** 2023-05-30

**Authors:** Olga Gruzdeva, Yulia Dyleva, Ekaterina Belik, Evgenia Uchasova, Anastasia Ponasenko, Sergey Ivanov, Maxim Zinets, Alexander Stasev, Anton Kutikhin, Victoria Markova, Alena Poddubnyak, Evgenia Gorbatovskaya, Elena Fanaskova, Olga Barbarash

**Affiliations:** 1Department of Experimental Medicine, Research Institute for Complex Issues of Cardiovascular Diseases, 6, Sosnovy Boulevard, 650002 Kemerovo, Russia; gruzov@kemcardio.ru (O.G.); dileua@kemcardio.ru (Y.D.); believ@kemcardio.ru (E.B.); uchaeg@kemcardio.ru (E.U.); ponaav@kemcardio.ru (A.P.); ivansv@kemcardio.ru (S.I.); zinemg@kemcardio.ru (M.Z.); stasan@kemcardio.ru (A.S.); markve@kemcardio.ru (V.M.); poddao@kemcardio.ru (A.P.); bichee@kemcardio.ru (E.G.); fanaev@kemcardio.ru (E.F.); barbol@kemcardio.ru (O.B.); 2Department of Pathophysiology, Kemerovo State Medical University, 650029 Kemerovo, Russia

**Keywords:** ceramides, serine palmitoyltransferase, ceramide synthase, sphingomyelinase, ceramidase, sphingomyelin synthase, epicardial adipose tissue, coronary artery disease

## Abstract

Here, we examined the expression of ceramide metabolism enzymes in the subcutaneous adipose tissue (SAT), epicardial adipose tissue (EAT) and perivascular adipose tissue (PVAT) of 30 patients with coronary artery disease (CAD) and 30 patients with valvular heart disease (VHD) by means of quantitative polymerase chain reaction and fluorescent Western blotting. The EAT of patients with CAD showed higher expression of the genes responsible for ceramide biosynthesis (*SPTLC1*, *SPTLC2*, *CERS1*, *5*, *6*, *DEGS1*, and *SMPD1*) and utilization (*ASAH1*, *SGMS1*). PVAT was characterized by higher mRNA levels of *CERS3*, *CERS4*, *DEGS1*, *SMPD1*, and ceramide utilization enzyme (*SGMS2*). In patients with VHD, there was a high *CERS4*, *DEGS1*, and *SGMS2* expression in the EAT and *CERS3* and *CERS4* expression in the PVAT. Among patients with CAD, the expression of *SPTLC1* in SAT and EAT, *SPTLC2* in EAT, *CERS2* in all studied AT, *CERS4* and *CERS5* in EAT, *DEGS1* in SAT and EAT, *ASAH1* in all studied AT, and *SGMS1* in EAT was higher than in those with VHD. Protein levels of ceramide-metabolizing enzymes were consistent with gene expression trends. The obtained results indicate an activation of ceramide synthesis de novo and from sphingomyelin in cardiovascular disease, mainly in EAT, that contributes to the accumulation of ceramides in this location.

## 1. Introduction

Albeit obesity is among the key modifiable risk factors in cardiovascular disease and adipose tissue (AT) represents an active endocrine organ intimately associated with the pathophysiology of coronary artery disease (CAD) [1], recent studies demonstrated a considerable heterogeneity of distinct fat depots [2]. As AT is unevenly distributed in the body and develops at different biomechanical and biochemical conditions, fat depots have significant differences in their cellular composition and molecular profile [2]. For instance, epicardial AT (EAT) is located in close proximity to the myocardium, and AT of the coronary arteries (termed further as perivascular AT, PVAT) is known to affect atherogenesis [3]. Hypertrophy, chronic low-grade inflammation, and dysfunction of EAT and PVAT are associated with the development of cardiovascular disease and major adverse cardiovascular events including sudden cardiovascular death, although subcutaneous AT (SAT) is less involved in cardiovascular pathology [3,4,5,6]. Nevertheless, SAT, EAT, and PVAT remain three the most frequent fat depots investigated in cardiovascular disease studies.

AT produces a wide range of biologically active substances, among which ceramides (sphingolipids) pose a special role. In addition to providing structural support to cell membranes, ceramides also play a crucial role in secondary messenger functions (participating in intracellular and intercellular signaling), regulating cell differentiation, proliferation, migration, apoptosis, and metabolism [7]. Epidemiological and experimental studies have demonstrated the relationship between ceramides and cardiovascular risk factors such as age, arterial hypertension, and obesity [8], and there is a pathogenetic link between altered ceramide profile and atherosclerotic progression [9]. Human plasma lipidome studies have identified certain types of ceramides acting as proatherogenic mediators which might serve as independent predictors of cardiovascular events [10]. However, due attention has not been given to ceramide accumulation in the EAT and PVAT of patients with CAD in relation to atherogenesis. Hence, here, we conceived the comparison of ceramide-metabolizing enzymes in the EAT and PVAT as compared to SAT located in the heart projection.

The biosynthesis of ceramides in cells is carried out in three ways, which are strictly localized in the cell. The main source of ceramides in adipocytes is the de novo pathway, which is realized in the endoplasmic reticulum [11]. Through the coordinated action of enzymes (serine palmitoyltransferase, 3-ketodihydrosphingosine reductase, ceramide synthase, dihydroceramide desaturase), up to 80% of cellular ceramides are generated as a result of four successive reactions. The presence of six isoforms of the ceramide synthase enzyme (CerS1-CerS6), which attaches the fatty acid acyl chain to the sphinganine backbone, ensures the high species diversity of ceramides [11]. CerS1, CerS5, and CerS6 add fatty acids with shorter hydrocarbon chain lengths (C14-C18) to sphinganine, while CerS2, CerS3, and CerS4 add longer-chain fatty acids (C18-C26 and longer) [12]. An imbalance of CerS in a cell will lead to an increase in certain ceramides, disrupting cell homeostasis [12]; however, the mechanisms of the disturbance of ratios and the significance of these ratios are not fully understood, and additional studies are needed. An alternative pathway for the formation of ceramides In cells is the sphingomyelinase pathway, which consists of the hydrolysis of sphingomyelin by sphingomyelinase (sMase) in several cell compartments, but this pathway is mainly realized in the plasma membrane. In lysosomes and endosomes, a pathway is implemented for the synthesis of ceramides via the cleavage of complex sphingolipids into sphingosine, which can be reused for ceramide resynthesis with the help of ceramide synthase [7].

To date, there are insufficient data regarding the expression of genes encoding enzymes for the synthesis of ceramides in AT, especially those localized in the cardiac system. Existing works are mainly experimental in nature, and the results are often contradictory [7,13]. A change in the activity of synthetic enzymes in heart AT can lead to excessive synthesis and accumulation of ceramides not only in the AT but also in cardiomyocytes. An excessive accumulation of ceramides, in turn, can induce a number of pathological processes associated with atherogenesis [14]. In this context, the aim of the study was to comparatively evaluate the expression of genes encoding enzymes of ceramide metabolism in the AT of the heart and blood vessels of patients with coronary artery disease and acquired heart defects.

## 2. Results

The inclusion criteria for patients in the comparison group were as follows: verified acquired heart disease; consented to the study. The clinical and anamnestic characteristics of patients are presented in Table 1. The patients in the study groups were comparable in terms of gender and age. Among patients with CAD, males predominated (Table 1).

In the anamnesis, risk factors for CVD were more often recorded. Patients with acquired heart defects were comparable in gender and age to patients in the main group. Patients with heart defects more often suffered from CHF II–III FC and had a reduced ejection fraction in comparison with the CAD group. All patients were administered standard antianginal and antiplatelet therapies throughout the observation period and in-hospital treatment period (Table 1).

### 2.1. Expression of Genes for De Novo Ceramide Biosynthesis Enzymes

To obtain information about the expression of genes encoding enzymes of the de novo ceramide synthesis pathway in AT, the mRNA levels of *SPT*, *CERS*, and *DEGS1* were assessed using quantitative real-time polymerase chain reaction. Taking into account that SPT consists of at least two subunits, the mRNA levels of *SPTLC1* and *SPTLC2* were assessed. Among patients with CAD, expression of the C1 subunit was found to be higher in SAT and EAT samples compared with AT of perivascular localization (*p* = 0.0002, *p* = 0.010, respectively) (Figure 1). In contrast to the C1 subunit, the expression level of the C2 subunit was higher only in EAT samples compared with SAT and PVAT (*p* = 0.012, *p* = 0.013, respectively).

Ceramide synthase gene expression in the AT of patients with CAD had tissue-specific features. SAT was characterized by maximum expression of the *CERS2* gene (Figure 1) encoding ceramide synthase 2, which is an enzyme producing long-chain ceramides and attaching C20–C24 fatty acyl CoA to the sphingoid base. The maximum expression of *CERS1*, a gene for an enzyme which attaches C18 fatty acyl CoA, *CERS4*, a gene for an enzyme which attaches C26–C34 fatty acyl CoA and produces very long-chain ceramides, as well as *CERS5* and *CERS6* genes encoding ceramide synthase with a specificity for C14–C16 fatty acyl CoA were found in EAT. Of note, *CERS5* expression was higher than the expression of *CERS6*. In contrast, PVAT was characterized by the pronounced expression of *CERS3*, which encodes ceramide synthases that attach C18–C20 and produce long-chain as well as *CERS4* genes and produce long-chain and very long-chain ceramides. The mRNA levels of *DEGS1*, the enzyme acting at the final stage of de novo ceramide synthesis, was higher in EAT and PVAT as compared with SAT (*p* = 0.010 and *p* = 0.012, respectively, Figure 1). In the group of patients with VHD, AT samples did not differ in terms of the mRNA levels of *SPTLC1*, *SPTLC2*, *CERS1*, *CERS2*, *CERS5*, and *CERS6*, while there was a high expression of *CERS3* in perivascular adipocytes (*p* = 0.004), and *CERS4* expression was notable in EAT (*p* = 0.011) and PVAT (*p* = 0.024) (Figure 1). The *DEGS1* expression in EAT samples was highest when compared with SAT and PVAT (*p* = 0.014 and *p* = 0.011, respectively) (Figure 1).

The evaluation of differences in the expression of ceramide synthesis enzymes in the AT of patients with CAD or VHD showed that patients with CAD, unlike patients with VHD, were characterized by higher *SPTLC1* expression in SAT and EAT samples (*p* = 0.00003, *p* = 0.0022, respectively) and higher *SPTLC2* in EAT samples (*p* = 0.039) (Figure 1). Among patients with CAD, there was also higher *CERS2* expression in the samples of all studied tissues (SAT, *p* = 0.0001, EAT, *p* = 0.003, PVAT, *p* = 0.0013) and higher *CERS4* and *CERS5* expression in EAT (*p* = 0.022, *p* = 0.017). No intergroup differences in *CERS6* gene expression were found (Figure 1). The mRNA levels of *DEGS1* among patients with CAD were higher in adipocytes regardless of their location (SAT, *p* = 0.029; EAT, *p* = 0.035; PVAT, *p* = 0.030) (Figure 1).

The determination of protein expression by Western blotting followed by semi-quantitative analysis using densitometry in ImageJ showed an increased expression of the SPTLC1 and DEGS1 proteins and weaker expression of the SPTLC2 and CERS6 proteins in samples of SAT, EAT, and PVAT (Figure 2, Appendix A). Full images of Western blotting gels are presented in the Appendix A.

### 2.2. Expression of Genes Encoding Enzymes of the Sphingomyelinase Pathway for Ceramide Synthesis

With the help of CERT transporter proteins, which work in an ATP-dependent manner, ceramides from the endoplasmic reticulum are transported to the Golgi apparatus, where they are metabolized to sphingomyelin by acidic or neutral sphingomyelinase (SMase) [15].

We found that the acid sphingomyelinase gene *SMPD1* was higher expressed than the neutral sphingomyelinase gene *SMPD3* in cardiac AT. At the same time, among patients with CAD, the mRNA levels of *SMPD1* were highest in EAT (*p* = 0.002) and SAT (*p* = 0.011) (Figure 3).

Among patients with VHD, the mRNA levels of *SMPD1* were higher in SAT than PVAT (*p* = 0.026). *SMPD3* expression did not show any tissue-specific features in either of the study groups (Figure 3). The presence of acidic or neutral sphingomyelinase (ASMase, NSMase2) proteins was recorded in AT samples from patients by Western blotting (Figure 2, Appendix A). Semi-quantitative analysis using densitometry also showed a higher expression of the ASMase protein compared to NSMase.

### 2.3. Gene Expression of Ceramide Degradation Enzymes

We also assessed the mRNA levels of genes for ceramide catabolism enzymes. With the help of the enzyme acid ceramidase (ASAH1), ceramide is deacylated to form sphingosine and free fatty acids. *ASAH1* gene expression in epicardial adipocytes of patients with CAD was maximal in comparison with adipocytes of other localizations (*p* = 0.015, *p* = 0.014) (Figure 4).

Patients with VHD were characterized by high *ASAH1* expression, which did not depend on the location of the AT. At the same time, in the group with heart defects, the levels of *ASAH1* mRNA were significantly higher in SAT (*p* = 0.0003), EAT (*p* = 0.037), and PVAT (*p* = 0.0021) compared with the CAD group. Protein expression of *ASAH1* was detected in all AT samples from patients with CAD and VHD (Figure 2, Appendix A).

Ceramides are also a substrate for the formation of a more complex sphingolipid, sphingomyelin. This reaction is catalyzed by the enzyme sphingomyelin synthase (SMS), of which there are several isoforms. In our study, we determined the expression of the SMS1 and SMS 2 (*SGMS1* and *SGMS2*) genes.

Patients with CAD were characterized by a higher level of *SGMS1* in epicardial adipocytes (*p* = 0.006 and *p* = 0.005 in SAT and PVAT, respectively), while the mRNA levels of *SGMS1* in the group of patients with VHD did not exhibit tissue specificity (Figure 4). *SGMS1* expression in the EAT was higher in patients with CAD than in patients with VHD (*p* = 0.0002). The expression of *SGMS2* was significantly higher than that of *SGMS1* in both study groups and was highest in adipocytes of SAT and PVAT compared with EAT (*p* = 0.0012, *p* = 0.0021 in the CAD group, respectively, and *p* = 0.0011, *p* = 0.0015, respectively, in the VHD group) (Figure 4). Patients with CAD were characterized by higher levels of *SGMS2* mRNA in subcutaneous (*p* = 0.029) and epicardial (*p* = 0.035) adipocytes. The levels of sphingomyelin synthase (SGMS1, SGMS2) proteins corresponded to the trends in gene expression (Figure 2, Appendix A).

## 3. Discussion

Experimental and clinical studies have shown an association between ceramides and the development of atherosclerosis [9]. AT can be a potential source of ceramides, since it has all the necessary substrates for their synthesis and, importantly, enzymatic systems for ceramide synthesis [11,16]. Of particular interest is the expression of enzymes for the synthesis of ceramides in the fat depots of the heart and blood vessels, since EAT and PVAT are localized in close proximity to the focus of atherosclerotic lesions.

The first stage of the study was to evaluate the expression of genes for de novo ceramide synthesis enzymes in the AT of subcutaneous, epicardial, and perivascular localization in coronary and non-coronary pathologies. It was found that in the EAT, the expression of *SPTLC1* and *SPTLC2* is higher in patients with CAD than in those with VHD. In the AT of subcutaneous, epicardial, and perivascular localization, the expression of all six ceramide synthase enzymes was found regardless of the presence of coronary or non-coronary nosology. The mRNA levels of the *CERS4* and *CERS2* genes were the highest, while the levels of *CERS6* were the lowest (regardless of AT location). The presence of the coronary pathology (CAD) was accompanied by the highest values of *CERS2* expression in SAT, EAT, and PVAT as well as the highest values of *CERS4* and 5 in EAT. A higher level of *DEGS1* mRNA was also found in the EAT in patients with CAD.

In CAD patients, an increased expression of de novo ceramide synthesis enzymes in SAT and EAT adipocytes may be due to various reasons, including being overweight, excess fatty acids, hypoxia, and inflammation [17]. Błachnio-Zabielska et al. demonstrated a significant increase in the mRNA of both SPT subunits (*SPTLC1* and *SPTLC2*) in abdominal SAT samples from obese individuals (BMI > 30) compared with non-obese controls (BMI < 25). Enzyme activity, assessed using the radiolabeled substrate [3H]-L-serine (Moravek Biochemicals), was also increased [18]. However, one should take into account the absence of differences in BMI in a group of patients with VHD who had lower *SPTLC1* expression values. On the other hand, myocardial ischemia/hypoxia due to the atherosclerosis of coronary arteries could also provoke an increase in the expression of enzymes of the de novo pathway in adipocytes. It is known that increased lipolysis due to activation of the sympathoadrenal system during myocardial ischemia/hypoxia is accompanied by the accumulation of saturated fatty acids in systemic circulation. Excess fatty acids are dangerous for the cell, since fatty acids are embedded in the elastic cell membrane, making it more rigid. Fatty acids strengthen the lipid framework of the membrane and create small dense islands on this framework, in which the usual movement of molecules slows down significantly; therefore, the elasticity of the membrane rapidly decreases, literally “hardening”, as a result of which the cell is gradually destroyed. Fatty acids can also affect the properties of membrane-bound receptors, signal transduction, and the activity of a large number of membrane-bound enzymes [19]. This is especially relevant and important for the endoplasmic reticulum, which is the largest membrane system and the center of lipid metabolism in a cell.

Thus, the synthesis of ceramides with the participation of fatty acids in adipocytes may initially be adaptive, but as ceramides accumulate in the cell, a slowdown in key cellular and physiological processes is observed and, ultimately, programmed cell death is activated. It is known that the rate of ceramide synthesis is controlled primarily by the presence of fatty acids, which are a substrate for SPT and are necessary for the formation of the sphingosine backbone. However, excess fatty acids can activate the Toll-like receptor (TLR)4 on adipocytes. The palmitate activation of TLR4 causes an accumulation of ceramides by activating the transcription factor nuclear factor kappa-light-chain-enhancer of activated B cells (NFκB) and increasing the expression of several enzymes of ceramide synthesis [20] in addition to inducing the synthesis of pro-inflammatory cytokines [16]. Pro-inflammatory cytokines, including tumor necrosis factor (TNF)-α and interleukin (IL)-1 and -6, in turn stimulate ceramide synthesis [21]. Experimental studies have shown that TNF-α stimulates the accumulation of ceramides in cultured cells by inducing the expression of de novo ceramide synthesis (SPT) genes and the expression of sphingomyelin hydrolyzing enzymes (sphingomyelinases) [22]. Thus, TLR4 activation alters the metabolic signaling program of adipocytes, converting fatty acids into sphingolipids (ceramides), which can disrupt the normal functioning of AT. It is believed that EAT has a greater ability to absorb and release free fatty acids [23], which may contribute to an increased expression of genes for ceramide biosynthesis enzymes in patients with CAD under conditions of inflammation/hypoxia/ischemia; the results of our study confirm this position.

An important aspect is also the hypoxia-induced increase in the pro-inflammatory activity of AT. It is believed that hypoxia that develops in areas of the ventricle leads to the production of pro-inflammatory TNF-α by adipocytes, which in turn enhances the expression of SPT [24]. Park et al. showed that the pharmacological inhibition of SPT leads to a decrease in plasma levels of sphingolipids and proatherogenic lipids and is associated with the regression of atherosclerotic lesions and the formation of a stable plaque phenotype. This means that the regulation of sphingolipid biosynthesis, particularly at the level of SPT, may have clinical applications in the treatment of atherosclerosis [9]. Ceramide synthases, which are encoded by the *CERS1–6* genes, harbor a substrate specificity for fatty acids with different chain lengths to ceramides. In mammals, the length of the acyl chain of ceramide may be medium (C12–C14), long chain (C16–C20), very long chain (C22–C26) or ultralong long chain (>C26).

It has been experimentally shown that the activation/blocking of ceramide synthases can have a wide range of functional and tissue-specific effects [25]. The results of our study indicate that all six ceramide synthase enzymes are expressed in the AT of subcutaneous, epicardial, and perivascular localization, regardless of the presence of coronary or non-coronary nosology. The expression of *CERS4* and *CERS2* genes was the highest, while the expression of *CERS6* was the lowest (regardless of AT location). The presence of coronary pathology (CAD) was accompanied by the highest values of *CERS2* expression in SAT, EAT, PVAT, while it was accompanied by the highest values of *CERS4* and *CERS5* expression in EAT. Currently available data from experimental and clinical studies do not allow unambiguous interpretation of the obtained results. Thus, we cannot draw an unambiguous conclusion about whether the ceramide synthases studied herein have a protective or negative role. According to Kim et al., an inhibition of *CERS4* expression improves the liver metabolic profile in mice [26]. Additionally, *CERS4* can generate C20 and C22 ceramides, which have a protective function in the development of heart failure [27]. *CERS2* synthesizes ceramides with a chain length of C22–C24. A distinctive feature of the *CERS2* gene is its organization, characteristic of a “housekeeping” gene, and its location in chromosomal regions that are replicated in the early stages of the cell cycle [28]. In the physiology of cells, including adipocytes, *CERS2* is of key importance, since the knockdown of *CERS2* leads to impaired cytokinesis [29]. Homozygous *CerS2* knockout mice are highly susceptible to maladaptive metabolic disorders when fed an obesity-inducing diet. Of note, *CerS2* overexpression is protective against palmitate-induced endoplasmic reticulum stress responses, presumably because it prevents CerS6 induction from leading to increased hepatic lipogenesis [26]. Other studies suggest a negative role for *Cers2*. Thus, the overexpression of *CerS2* (and accumulation of C20-C24 ceramides) causes oxidative stress and mitochondrial dysfunction through lipid overload, which ultimately leads to cardiomyocyte apoptosis [30]. Some authors believe that *CerS2*-dependent damage to cell mitochondria may be a combination of other pathological conditions such as insulin resistance, oxidative stress, and increased autophagy and mitophagy, constituting a pathophysiological mechanism leading to the death of cardiomyocytes. It has also been shown that *CerS2* synergistically enhances the expression of *CerS5*, which was clearly shown to be involved in the induction of oxidative stress and apoptosis [30]. Mice with null *CerS5* are viable, fertile, and do not have any obvious morphological and phenotypic changes with normal nutrition. However, on a high-fat diet, the loss of *CerS5* is associated with reduced weight gain, improved general condition, reduced inflammatory activation of white AT, and decreased leptin levels compared with wild-type animals [31].

The last stage of the de novo pathway is considered equally important, since it is ceramides and not dihydroceramides that are the end products with differing metabolic and cellular activity profiles. The obtained results indicate a higher mRNA level of *DEGS1* in EAT in patients with CAD. Such differences have not been found in previous studies; moreover, high *DEGS1* expression in subcutaneous adipocytes compared with adipocytes of the mediastinal depot was noted in overweight patients referred for elective surgery on the aortic valve and/or on the ascending thoracic aorta [32]. It is noteworthy that in patients with CAD, the mRNA levels of the *DEGS1* in the SAT were significantly lower than the mRNA values of the first enzyme of the de novo SPT pathway. On the contrary, the values of *DEGS1* expression in EAT corresponded with SPT enzyme levels. A possible explanation for the overexpression of desaturase in EAT is the leveling of the pathological effects of hypoxia. Some researchers believe *DEGS1* functions in cellular oxygen sensors [33].

An alternative pathway for the formation of ceramides in the cell is the degradation of sphingomyelin under the action of sphingomyelinases (SMase), resulting in the formation of ceramide and phosphocholine [34]. The reaction occurs predominantly in the plasma membrane, but enzymes have also been identified in lysosomes and the endoplasmic reticulum. The formation of ceramides through this path occurs much faster, in contrast to de novo synthesis, as only one chemical reaction is involved. There are three main types of SMase according to their optimum pH. These are lysosomal and plasmatic acidic SMase (ASMase), neutral endoplasmic reticulum/nucleus and plasma membrane SMase (NSMase), and alkaline SMase (alk-SMase), which is present in the human intestinal tract and bile and was therefore not evaluated in this study [34]. ASMase enzyme activity is found in all heart tissues of rats, mice, and humans [35] and functions as a housekeeping gene in lysosomes. It has been experimentally shown that the administration of an ASMase inhibitor prior to ischemic injury reduces ischemia-induced cell death [36]. In ischemia/reperfusion models of mice, rabbits, and rats, an increase in ceramide, ASMase, and NSMase levels in the area of myocardial infarction and an association with destabilization of atherosclerotic plaques among patients with acute coronary syndrome has been demonstrated [37]. According to our data, the expression of the ASMase (*SMPD1*) gene in the EAT of patients with CAD is more pronounced than for other ATs and patients with VHD; therefore, it can be assumed that this pathway is also activated in EAT, and ASMase synthesized in AT may be involved in atherogenesis. The results obtained are consistent with earlier studies by Kolak and colleagues, who showed an increased expression of sphingomyelinase in visceral AT compared to SAT among obese patients [32].

It has been shown that high levels of pro-inflammatory cytokines are associated with atherosclerosis. TNF-α, IL-1β and interferon-γ stimulate SMase activity in the endothelial cells of vessels affected by atherosclerosis, thereby increasing the concentration of ceramides in cells [38], which probably occurs in adipocytes of AT of the heart and coronary arteries. Previously, we showed there is a high expression of pro-inflammatory adipocytokines in EAT [3]. Increased ceramide synthesis via ASMase activation may be related to TLR4 activation [39], since it has been shown that TLR4 activation by exogenous C2 ceramide induces ceramide synthesis via not only the de novo pathway but also the sphingomyelinase pathway [39]. Along with the activation of inflammation in the EAT, we also found an increase in the size of the fat depot of this area in patients with CAD [3], which, against the background of dyslipidemia, can lead to local hypoxia and the activation of factors induced by hypoxia such as, for example, HIFs. HIF-1α in adipocytes in turn increases the expression of SMase, thereby activating the synthesis of ceramides from sphingomyelin [40]. The authors also showed that this pathway is associated with the development and progression of atherosclerosis.

To maintain cellular homeostasis and the functional activity of cells, including adipocytes, a balance of synthesis enzymes and ceramide utilization enzymes is necessary. A deficiency of catabolism enzymes can lead to the accumulation of ceramides in the cell. Ceramidase enzymes deacylate ceramides to form sphingosine and free fatty acids, which are classified according to their optimum pH. In humans, five types of ceramidases are known: acidic, neutral and alkaline ceramidases 1, 2, and 3, which are encoded by five different genes (*ASAH1*, *ASAH2*, *ACER1*, *ACER2*, and *ACER3*, respectively). The expression of these genes is tissue-specific; *ASAH1* is expressed in heart tissues [41]. *ASAH1* activity in lysosomes is modulated by the protein saponin D, the deficiency of which leads to the accumulation of ceramides in tissues [41]. According to our data, *ASAH1* expression among patients with CAD was maximal in EAT, while the mRNA level of the *ASAH1* was significantly higher in EAT and PVAT samples from patients with VHD.

The level of ceramidase expression was significantly higher in the adipocytes of the heart and blood vessels of patients with VHD. Therefore, it should be assumed that along with the high activity of ceramide synthesis enzymes in non-coronary heart pathology, ceramide-utilizing enzymes are also highly expressed, in particular lysosomal ceramidase, which deacylates ceramides produced from the degradation of plasma membrane sphingolipids. Asah1 activity has been shown to be higher with respect to unsaturated short- and long-chain ceramides [42]. On the other hand, this pathway of ceramide catabolism is also present in the EAT of patients with CAD, although *ASAH1* expression is significantly lower in comparison to patients with VHD. Sphingosine obtained as a result of this reaction can, on the one hand, be phosphorylated by sphingosine kinase to sphingosine-1-phosphate, which has the opposite properties to ceramides and, on the other hand, it can be a substrate for CERS and be re-included in the synthesis of ceramides by the de novo pathway [14].

Another enzyme that catabolizes ceramides is sphingomyelin synthase, which has two isoforms (SMS1 and SMS2) that, in the course of a chemical reaction, form sphingomyelin from ceramides [43]. Ceramides synthesized in the endoplasmic reticulum are transported via vesicular and non-vesicular transport to the Golgi apparatus, where they are metabolized into complex sphingolipids, including SM [14,42,44]. It has been established that SMS1 is localized in the Golgi apparatus, while SMS2 is predominantly localized in the plasma membrane [42]. According to our data, the maximum *SGMS1* expression in the EAT combined with the maximum *SGMS2* expression in the SAT and PVAT was higher in patients with CAD than in patients with VHD. Based on this, it can be assumed that in coronary pathology, sphingomyelin synthesis pathways are activated not only inside adipocytes with the activation of de novo ceramide synthesis but also, to a greater extent, in the plasmatic membrane of adipocytes. Sphingomyelin has been shown to be the main sphingolipid in atherogenic apoB-containing lipoproteins, including low and very low-density lipoproteins and chylomicrons. Numerous instances of data indicate that the content of sphingomyelin in the aortic wall and in plasma is associated with atherosclerosis. Sphingomyelin accumulates in atheromas in humans, and this has also been confirmed in animal models. Human atherosclerotic plaque low-density lipoproteins contain much more sphingomyelin than plasma low-density lipoproteins [45]. The level of sphingomyelin in human plasma is an independent risk factor for CAD [46] and is also associated with an increased risk of myocardial infarction [47]. A combination of SMS1+2 deficiency with ApoE or low-density lipoproteins receptor knockout is associated with reduced atherosclerotic manifestations [45].

## 4. Materials and Methods

### 4.1. Study Population

This study included 60 patients: 30 patients with CAD (main group) and 30 patients with acquired degenerative non-rheumatic valvular heart disease (stenosis/insufficiency aortic valve) (VHD). All the patients had indications for open heart intervention: either direct myocardial revascularization by coronary bypass grafting (CABG) or heart valve surgery.

The inclusion criteria for patients were as follows: age up to 75 years; consented to the study. The exclusion criteria for patients were as follows: age over 75 years; the presence of diabetes mellitus type 1 and 2 in history and/or detected during the examination during the period of hospitalization; myocardial infarction; clinically significant concomitant pathology (anemia, renal, or hepatic failure, oncological and infectious inflammatory diseases during exacerbation, autoimmune diseases); refusal of the patient to consent to the study.

### 4.2. Sample Collection

Biopsy samples of AT of subcutaneous, epicardial and perivascular localization (3–5 g of fat) were obtained from patients during surgery (CABG or correction of heart defects). The source of SAT was the subcutaneous tissue of the lower angle of the mediastinal wound; the source of EAT was the area of its greatest presence in the right parts of the heart (right atrium and right ventricle); the source of PVAT was the region of the right coronary artery. AT samples were subjected to cryogenic freezing with liquid nitrogen followed by storage at a temperature of −150 °C.

### 4.3. RNA Extraction and Real-Time Quantitative Polymerase Chain Reaction (RT-qPCR)

The isolation of RNA from AT was performed using the Fatty Tissue RNA Purification Kit (Norgen Biotek Corp., Thorold, ON, Canada), the main advantages of which are rapid isolation and the high degree of purity of total RNA from lipid-rich tissues. The amount and purity of the isolated RNA were assessed using a NanoDrop 2000 Spectrophotometer (Thermo Scientific, Pleasanton, CA, USA). For reverse transcription and the synthesis of complementary DNA (cDNA) based on RNA samples, the High-Capacity cDNA Reverse Transcription Kit with RNase inhibitor reagents was used (Applied Biosystems, Foster City, CA, USA). Samples were stored at –20 °C for two days until qPCR was performed.

The expression of the genes encoding enzymes for the synthesis and degradation of ceramides was evaluated by quantitative real-time polymerase chain reaction (qPCR) with primers synthesized by Evrogen (Moscow, Russia) on a ViiA 7 Real-Time PCR System (Applied Biosystems, Foster City, CA, USA). The primer sequences are presented in Table 2.

To perform PCR with the intercalating dye SYBR Green, reaction mixtures were prepared using master-mix BioMaster UDG HS-qPCR Lo-ROX SYBR (2×) (Biolabmix, Novosibirsk, Russia) according to the manufacturer’s protocol. Quantitative PCR was performed using a CFX-96 Real-Time System amplifier (Bio-Rad, Hercules, CA, USA). The results were normalized using the reference genes ACTB (β-actin), GAPDH (glyceraldehyde-3-phosphate dehydrogenase), and B2M (beta-2-microglobulin). To calculate the relative value of expression, the ∆CT method (a variant of the Livak method) was used, which is based on determining the difference between the CT values of the reference genes and the target CT values for each sample and is represented on a logarithmic (log10) scale as the fold change relative to control samples. We used the geometric mean to accurate averaging the reference genes.

### 4.4. Fluorescent Western Blotting

For protein analysis, samples of subcutaneous, epicardial and perivascular adipose tissue of patients with CAD and VHD (*n* = 3–6 in the group of CAD and *n* = 3 in the group of VHD) were washed with a physiological saline (Hematek, Tver, Russia) and homogenized (FastPrep-24 5G, MP Biomedicals, San Diego, CA, USA; Lysing Matrix S Tubes, 116925050-CF, MP Biomedicals, San Diego, CA, USA) in T-PER buffer (78510, Thermo Fisher Scientific, Waltham, MA, USA) supplied with the Halt protease and phosphatase inhibitor cocktail (78444, Thermo Fisher Scientific, Waltham, MA, USA) according to the manufacturer’s protocol. Upon the initial centrifugation at 14,000× *g* (Microfuge 20R, Beckman Coulter, Brea, CA, USA) for 10 min, supernatant was additionally centrifuged at 200,000× *g* (Optima MAX-XP, Beckman Coulter, Brea, CA, USA) for 1 h to sediment insoluble ECM proteins. Quantification of total protein was conducted using a BCA Protein Assay Kit (23227, Thermo Fisher Scientific, Waltham, MA, USA) and Multiskan Sky microplate spectrophotometer (Thermo Fisher Scientific, Waltham, MA, USA) in accordance with the manufacturer’s protocol.

Equal amounts of protein (20 μg per sample) were mixed with OrangeMark sample buffer (K-023, Molecular Wings, Kemerovo, Russia) in a 6:1 ratio, denatured at 99 °C for 5 min, and then loaded on a 15-well 1.5 mm NuPAGE 4–12% Bis-Tris protein gel (NP0336BOX, Thermo Fisher Scientific, Waltham, MA, USA). A Chameleon Duo Pre-stained Protein Ladder (928-60000, LI-COR Biosciences, Lincoln, NE, USA) was loaded as a molecular weight marker. Proteins were separated by the sodium dodecyl sulfate-polyacrylamide gel electrophoresis (SDS-PAGE) at 150 V for 2 h using a G-RUN MES running buffer (K-021, Molecular Wings, Kemerovo, Russia), G-NOOOX antioxidant (K-027, Molecular Wings, Kemerovo, Russia), and XCell SureLock Mini-Cell vertical mini-protein gel electrophoresis system (EI0001, Thermo Fisher Scientific, Waltham, MA, USA). Protein transfer was performed using nitrocellulose transfer stacks (IB23001, Thermo Fisher Scientific, Waltham, MA, USA) and an iBlot 2 Gel Transfer Device (Thermo Fisher Scientific, Waltham, MA, USA) according to the manufacturer’s protocols using a standard transfer mode for 30–150 kDa proteins (P0—20 V for 1 min, 23 V for 4 min, and 25 V for 2 min). Nitrocellulose membranes were then incubated in Block’n’Boost! solution (K-028, Molecular Wings, Kemerovo, Russia) for 1 h to prevent non-specific binding. Blots were probed with rabbit antibodies to SPTL (SPTLC1 antibody, CSB-PA022639ESR1HU, SPTLC2 antibody, CSB-PA022640LA01HU, Cusabio, Wuhan, China), CERS6 (CERS6 antibody, CSB-PA751145LA01HU, Cusabio, Wuhan, China), DEGS1 (DEGS1 antibody, CSB-PA617990, Cusabio, China), SMPD1 (aSMAse) (ASManti-SMPD1 antibody, FNab08040, FineTest, Wuhan, China), SMPD2 (nSMAse) (SMPD2 antibody, CSB-PA021846LA01HU, Cusabio, Wuhan, China), SMPD3 (nSMAse) (SMPD3 antibody, CSB-PA878920LA01HU, Cusabio, Wuhan, China), ASAH1 (ASAH1 antibody, CSB-PA619774FSRIHU, Cusabio, Wuhan, China), SMS1 (SGMS1 antibody, CSB-PA801243LA01HU, Cusabio, Wuhan, China), SMS2 (SGMS2 antibody, CSB-PA854159LA01HU, Cusabio, Wuhan, China) IRDye 680RD-conjugated goat anti-rabbit IgG secondary antibody (926-32211, LI-COR Biosciences, Lincoln, NE, USA) was used at 1:1000 dilution, respectively. Incubation with the antibodies was performed using Block’n’Boost! solution (K-028, Molecular Wings, Kemerovo, Russia), iBind Flex Cards (SLF2010, Thermo Fisher Scientific, Waltham, MA, USA) and Bind Flex Western Device (Thermo Fisher Scientific, Waltham, MA, USA) according to the manufacturer’s protocols.

Fluorescent detection was performed using an Odyssey XF imaging system (LI-COR Biosciences, Lincoln, NE, USA) at a 800 nm channel (785 nm excitation and 830 nm emission). Total protein normalization was conducted after the fluorescent detection using 0.1% Fast Green FCF (F8130, Solarbio Life Sciences, Beijing, China) dissolved in 30% methanol (8.05.00186, ChemExpress, Ufa, Russia) and 7% acetic acid (61–75, EKOS-1, Moscow, Russia). Staining of the nitrocellulose membrane with 0.1% Fast Green FCF (F8130, Solarbio Life Sciences, Beijing, China) for 10 min was followed by destaining in 30% methanol (8.05.00186, ChemExpress, Ufa, Russia) and 7% acetic acid (61–75, EKOS-1, Moscow, Russia) for another 10 min and in two changes of deionized water (2 min per each). Total protein visualization was performed using an Odyssey XF imaging system (LI-COR Biosciences, Lincoln, NE, USA) at a 600 nm channel (520 nm excitation and 600 nm emission). Densitometry was performed using the ImageJ software (1.53t version, National Institutes of Health, Bethesda, MD, USA) using the standard algorithm (consecutive selection and plotting of the lanes with the measurement of the peak area) and subsequent adjustment of the specific signal from the antibody to the level of total protein transferred to a nitrocellulose membrane. The protocol of band densitometry and adjustment to the total protein loading was as follows: (1) analysis of the Western blot, ImageJ command chain: select first lane—select next lane—plot all lanes as distinct peaks—measurement of the peak area; (2) analysis of the total protein staining performed after the Western blot, the same ImageJ command chain; (3) calculation of the adjustment coefficient for each lane by the division of the highest total sum of total protein staining peak area on the membrane by the total sum of total protein staining peak area for every other lane on the membrane; (4) multiplication of the specific staining peak area for each lane by the respective adjustment coefficient to normalize specific staining intensity for the total protein loading. Total protein-adjusted specific staining values were used for the statistical analysis. Each Western blot underwent total protein normalization, i.e., specific staining was adjusted to the total protein staining.

### 4.5. Statistical Analysis

Statistical analysis was performed using GraphPad Prism 8 (GraphPad Software, San Diego, CA, USA). Data are presented as median and interquartile range (25th and 75th percentiles). A comparison of three independent groups was performed using the Kruskal–Wallis test followed by pairwise comparison using the Mann–Whitney U-test. A comparison of two independent groups was performed using the Mann–Whitney U-test. Categorical variables, expressed as percentages, were compared using the Pearson’s chi-squared test or Fisher’s exact test. *p*-values < 0.05 was considered statistically significant.

## 5. Conclusions

Heart AT depots (mainly EAT) in patients with CAD are characterized by an increased expression of enzymes responsible for the de novo synthesis of ceramides and those participating in the sphingomyelin degradation pathway, which together contribute to the accumulation of ceramides and potentiating cardiovascular pathology. The observed increase may be at least partially explained by an excess of fatty acids, hypoxia, inflammation, and activation of TLR4 on adipocytes. In addition to an increased expression of ceramide biosynthetic enzymes, the EAT of patients with CAD also showed an elevated expression of enzymes catabolizing ceramides. We suggest that both the anabolism and catabolism of ceramides can be regulated in therapeutic context, most likely by pharmacological inhibition. Ceramide degradation products (for example, sphingosine) may serve as substrates for the production of compounds that have opposite effects to “bad” ceramides (e.g., sphingosine-1-phosphate) but are also substrates for the repeated synthesis of ceramides such as sphingomyelin (which, however, is itself a substrate for the synthesis of sphingomyelin-1-phosphate, which counteracts the pathological effects of ceramides). Importantly, many ceramides are indispensable for cell physiology. Therefore, a fine-tuned control of ceramide metabolism might be a promising strategy in the struggle against cardiovascular diseases, although it requires finding which ceramides and which of their degradation products should be blocked or activated.

### Study Limitations

We note that our study has certain limitations. First, it is a single-center study and, second, the sample size is small. Thirdly, the limitation is the lack of a healthy individual in the comparison. Fourth, lipidomic profiling of the fat deposits of the heart and coronary vessels of patients with cardiovascular diseases is needed, which is part of the plan of future work.

## Figures and Tables

**Figure 1 ijms-24-09494-f001:**
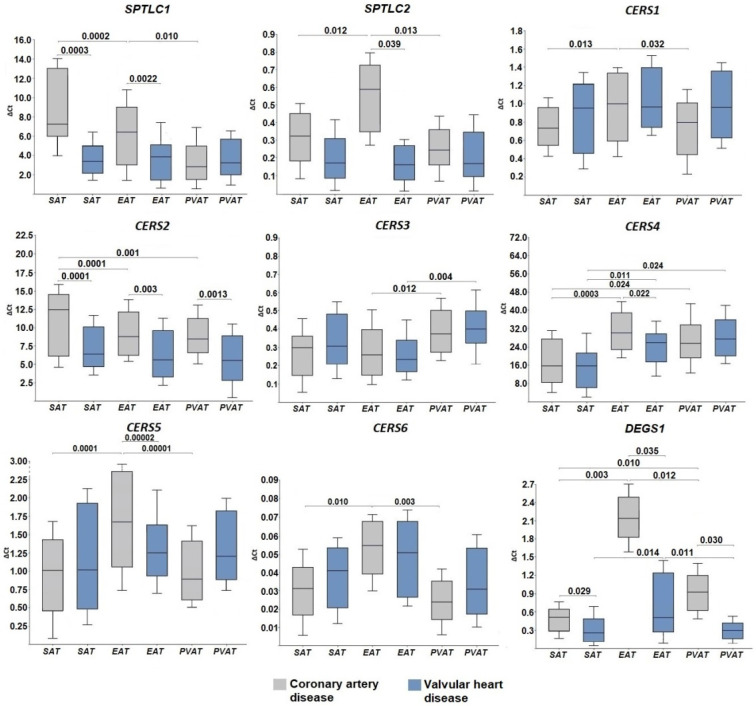
Expression of genes for de novo ceramide synthesis enzymes in the adipose tissue of the heart from various locations in patients with cardiovascular diseases. Data are presented in Me (Q1; Q3); *SPTLC1*, serine palmitoyltransferase subunit C1 gene; *SPTLC2*, serine palmitoyltransferase subunit C2 gene; *CERS1*, *CERS2*, *CERS3*, *CERS4*, *CERS5*, *CERS6*, ceramide synthase 1–6 gene expression; *DEGS1*, dihydroceramide desaturase 1 gene; SAT, subcutaneous adipose tissue; EAT, epicardial adipose tissue; PVAT, perivascular adipose tissue.

**Figure 2 ijms-24-09494-f002:**
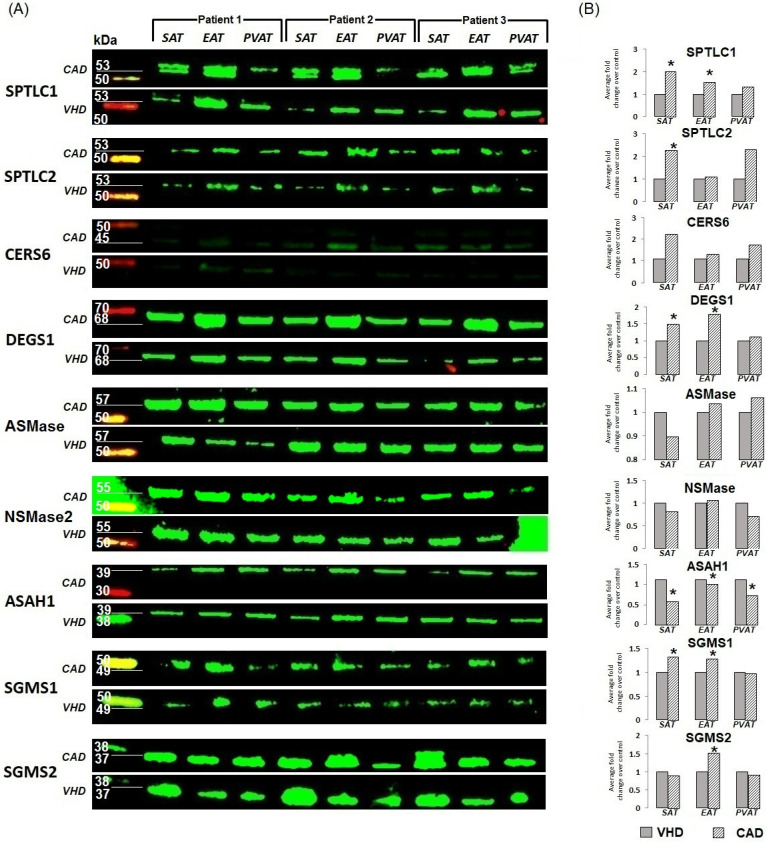
Changes in ceramide enzyme expression versus housekeeping protein expression in heart and vascular adipose tissue of patients with cardiovascular diseases. (**A**) Gel photo of Western blots for each protein quantified; (**B**) Semi-quantitative densitometry analysis performed after the pairwise adjustment of the specific signal for total protein staining. Data are presented as fold change relative to controls (the group of patients with VHD was taken as a unit). * *p* < 0.05. SAT, subcutaneous adipose tissue; EAT, epicardial adipose tissue; PVAT, perivascular adipose tissue; CAD, coronary artery disease; VHD, valvular heart disease; SPTLC1, serine palmitoyltransferase subunit C1; SPTLC2, serine palmitoyltransferase subunit C2; CERS1-6, ceramide synthase 1–6; DEGS1, dehydroceramide desaturase 1; SMPD1, acidic sphingomyelinase; SMPD2, neutral sphingomyelinase; ASAH1, acid ceramidase; SGMS1, sphingomyelin synthase 1; SGMS2, sphingomyelin synthase 2.

**Figure 3 ijms-24-09494-f003:**
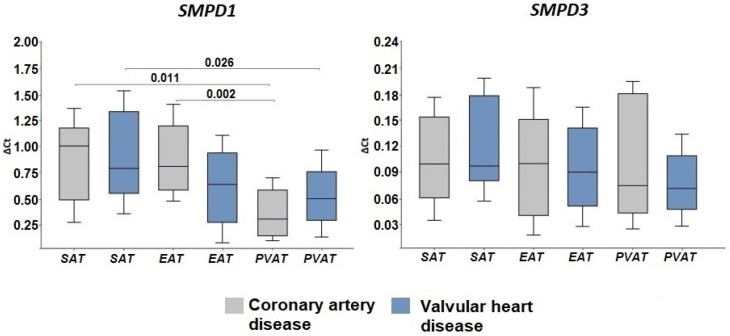
Expression of genes ceramide synthesis enzymes from sphingomyelin in the adipose tissue of the heart from various locations in patients with cardiovascular diseases. Data are presented in Me (Q1; Q3); *SMPD1*, acid sphingomyelinase genes; *SMPD3*, neutral sphingomyelinase genes; SAT, subcutaneous adipose tissue; EAT, epicardial adipose tissue; PVAT, perivascular adipose tissue.

**Figure 4 ijms-24-09494-f004:**
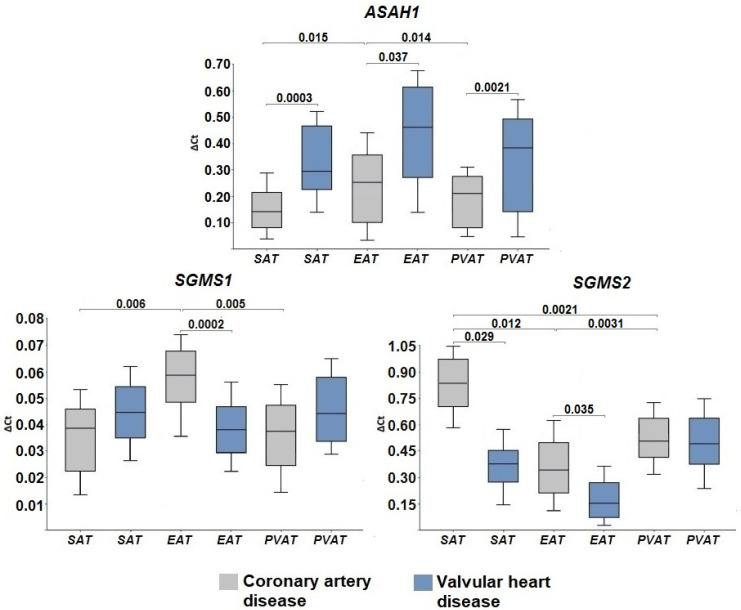
Expression of genes for ceramide degradation enzymes in the adipose tissue of various fat depots of the heart of cardiovascular disease patients. Data are presented in *n* (%) or Me (Q1; Q3); CAD, coronary artery disease; VHD, valvular heart disease; SAT, subcutaneous adipose tissue; EAT, epicardial adipose tissue; PVAT, perivascular adipose tissue; *ASAH1*, acid ceramidase gene expression; *SGMS1*, sphingomyelin synthase 1 gene expression; *SGMS2*, sphingomyelin synthase 2 gene expression.

**Table 1 ijms-24-09494-t001:** Clinical and anamnestic characteristics of patients.

Parameter	Patients with CAD, *n* = 30	Patients with VHD, *n* = 30	*p*
Male gender, *n* (%)	18 (60.0)	17 (56.7)	0.056
Age, median (range), years	64.9 (47.8; 69.5)	59.3 (43.7; 62.1)	0.071
Body mass index, median (range), kg/m^2^	26.4 (22.5; 30.2)	27.3 (23.4; 31.2)	0.062
Arterial hypertension, *n* (%)	17 (56.7)	7 (23.3)	0.002
Hypercholesterolemia, *n* (%)	13 (43.3)	3 (10.0)	0.001
Smoking, *n* (%)	15 (50.0)	5 (16.7)	0.0001
Family history of CAD, *n* (%)	18 (60.0)	11 (36.7)	0.038
Past medical history of myocardial infarction, *n* (%)	21 (70.0)	0	
Past medical history of stroke, *n* (%)	3 (10.0)	0	
Extracoronary atherosclerosis, *n* (%)	5 (16.7)	0	
No chronic coronary syndrome, *n* (%)	1 (3.3)	30 (100.0)	0.0001
Chronic coronary syndrome, FC I, *n* (%)	0	0	
Chronic coronary syndrome, FC II, *n* (%)	14 (46.7)	0	
Chronic coronary syndrome, FC III, *n* (%)	15 (50.0)	0	
Chronic coronary syndrome, FC IV, *n* (%)	0	0	
Chronic heart failure, NYHA I FC, *n* (%)	12 (40.0)	7 (23.3)	0.055
Chronic heart failure, CHF NYHA II FC, *n* (%)	4 (13.3)	23 (76.7)	0.002
Chronic heart failure, NYHA III FC, *n* (%)	0	0	
Chronic heart failure, NYHA IV FC, *n* (%)	0	0	
Atherosclerosis of 1 coronary artery, *n* (%)	4 (13.3)	0	
Atherosclerosis of 2 coronary arteries, *n* (%)	1 (3.3)	0	
Atherosclerosis of ≥3 coronary arteries, *n* (%)	24 (80.0)	0	
Ejection fraction, median (range), %	53.6 (46.3; 58.9)	51.6 (42.5; 55.8)	0.046
Treatment strategy (hospital period)			
Aspirin, *n* (%)	28 (93.3)	0	
Clopidogrel, *n* (%)	4 (13.3)	0	
Warfarin, *n* (%)	0	25 (83.3)	
β-blockers, *n* (%)	27 (90.0)	26 (86.7)	0.091
Angiotensin-converting enzyme inhibitors, *n* (%)	23 (76.7)	24 (80.0)	0.247
Statins, *n* (%)	30 (100.0)	22 (73.3)	0.059
Calcium channel blockers, *n* (%)	23 (76.7)	21 (70.0)	0.166
Nitrates, *n* (%)	1 (3.3)	2 (6.7)	0.107
Diuretics, *n* (%)	24 (80.0)	25 (83.3)	0.087

Data are presented in *n* (%) or Me (Q1; Q3); P, level of statistical significance; CAD, coronary artery disease; VHD, valvular heart disease; FC, functional class.

**Table 2 ijms-24-09494-t002:** Nucleotide sequence of primers used to assess expression of the genes of interest.

Gene	Orientation	Sequence (5′ -> 3′)	Primer Length
Serine palmitoyltransferase subunit C1 gene (*SPTLC1*)	Forward primer	aggaagcggctaactatggc	20
	Reverse primer	ccagaggatcagaatcccttcc	22
Serine palmitoyltransferase subunit C1 gene (*SPTLC2*)	Forward primer	cgcctgaaagagatgggcttc	21
	Reverse primer	ccgatgttccgcttcagcat	20
Ceramide synthase 1 genes (*CERS1*)	Forward primer	gcgtttgcagccaaggtgtt	20
	Reverse primer	ttcaccaggccgttcctcag	20
Ceramide synthase 2 genes (*CERS2*)	Forward primer	ggacgtgtctacgccaaagc	20
	Reverse primer	atgttcaagagggcagccagt	21
Ceramide synthase 3 genes (*CERS3*)	Forward primer	ctcgcacagatggtgtcctg	20
	Reverse primer	cctgatgggatgttgcttcctg	22
Ceramide synthase 4 genes (*CERS4*)	Forward primer	caggacttgttggcagccct	20
	Reverse primer	cgttgggcttcacttgcctc	20
Ceramide synthase 5 genes (*CERS5*)	Forward primer	ctcaatggcctgctgctgac	20
	Reverse primer	tgctctccacatcactgcga	20
Ceramide synthase 6 genes (*CERS6*)	Forward primer	cggacctgaagaacacggagga	22
	Reverse primer	atggcgcacggtttggctac	20
Dihydroceramide desaturase 1 gene (*DEGS1*)	Forward primer	ccactgagctggagtttcct	20
	Reverse primer	caggaattgtagtgagggaggt	22
Acid sphingomyelinase gene (*SMPD1*)	Forward primer	ccgctggctctatgaagcgat	21
	Reverse primer	cggggtatggggaaagagcat	21
Neutral sphingomyelinase gene (*SMPD3*)	Forward primer	gaaggacaacaaggtcccagt	21
	Reverse primer	caactccggctggtcaatgg	20
Acid ceramidase gene (*ASAH1*)	Forward primer	ctgaaccgcaccagccaaga	20
	Reverse primer	ggcagtcccgcaggtaagttt	21
Sphingomyelin synthase 1 gene (*SGMS1*)	Forward primer	ccagtgcaacgtgacgacag	20
	Reverse primer	agtccacactccttcagtcgct	22
Sphingomyelin synthase 2 gene (*SGMS2*)	ccgctggctctatgaagcgat	actctacctgtgcctggaatgc	22
	cggggtatggggaaagagcat		
	Reverse primer	tcagtgtcagcgtaaccgtgt	21
Beta-actin (*ACTB)*	Forward primer	catcgagcacggcatcgtca	20
	Reverse primer	tagcacagcctggacagcaac	21
Glyceraldehyde-3-phosphate dehydrogenase *(GAPDH)*	Forward primer	agccacatcgctcagacac	19
	Reverse primer	gcccaatacgaccaaatcc	19
Beta-2-microglobulin (*B2M)*	Forward primer	tccatccgacattgaagttg	20
	Reverse primer	cggcaggcatactcatctt	19

## Data Availability

The data presented in this study are available in the article or its Appendix A. Other data presented in this study are available on request from the corresponding author.

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
