# Peer review of "Expression of Ceramide-Metabolizing Enzymes in the Heart Adipose Tissue of Cardiovascular Disease Patients"

_ijms, 2023, doi:10.3390/ijms24119494_

Round 1
Reviewer 1 Report
Dear Gruzdeva et al.,
Thank you for submitting your manuscript entitled “Expression of ceramide-metabolising enzymes in the heart ad-2 ipose tissue of cardiovascular disease patients” to “International Journal of Molecular Sciences”. After carefully reviewing your submission, I have the following comments and recommendations:
Major concern:
· The study doesn't just concentrate on adipose tissue in the epicardium. So why is the title of the manuscript misleading?
· The paper does not provide a clear justification for why adipose tissues from various areas were studied. And it is not adequately addressed why the expression of ceramide-metabolism enzymes varies among adipose tissues of different origin.
· The western blotting image (Figure 2A) does not show the size of the proteins. Additionally, the control's western blotting data is absent, which calls into question the accuracy of the densitometric comparison (Figure 2B). How have the authors normalized the western blotting data needs to be explained better.
· mRNA expression data of SMPD2 and protein expression data of SMPD3 are missing.
· Statistical analysis method is missing.
· It is still unclear why Table 2 was included rather than Figure. The meaning of P1.2, P1.3, P3.5 etc. in table 2 is unclear.
· Three reference genes (ACTB, GAPDH and B2M) have been mentioned on the method section. The authors need to clarify if they use all 3 reference genes or single reference gene to calculate relative value of mRNA expression of all target genes.
· On the conclusion section, the authors have mentioned that ceramide metabolism can be controlled by regulating the expression of ceramide synthesis and degradation enzymes. But the study showed that both ceramide synthesis and degradation enzymes are upregulated, which doesn’t correlate with the mentioned sentence of conclusion.
· The study lacks control subjects/healthy individual.
Minor concern:
· Please rewrite the sentences on lines 115-120: “SAT was characterized by maximum……. and C26, was revealed”.
· The manuscript is missing table 4, which is mentioned in sub-section 2.3 (Gene expression of ceramide degradation enzymes). Please, revise the section.
· Table 3 didn’t mention the primer pairs of reference genes (ACTB, GAPDH and B2M).
· Please recheck the mentioned centrifugal speed (200,000 × g ) on line 473.
· On line 518, it was mentioned: “Total protein visualisation was performed using…… ”. The image of total protein is missing. If the authors used total protein for normalization, that should be clearly mentioned.
Overall, I recommend making the revisions outlined above before the manuscript can be considered for publication.
Thank you for considering my comments and recommendations, and I look forward to seeing a revised version of your manuscript.
Author Response
We sincerely thank the reviewer for the constructive criticism and valuable notes, which collectively helped us to improve the paper. Please see the attachment.

Reviewer 2 Report
The authors describe an interesting study regarding the the expression of genes encoding enzymes of ceramide metabolism in the AT of the heart and blood vessels of patients with coronary artery disease and acquired heart defects. The authors results show that EAT of patients with CAD showed higher expression of the genes responsible for ceramide biosynthesis (SPTLC1, SPTLC2, CERS1, 5, 6, DEGS1, and SMPD1) and utilization (ASAH1, SGMS1). PVAT was characterized by higher mRNA levels of CERS3, CERS4, DEGS1, SMPD1, and ceramide utilization enzyme (SGMS2). In patients with VHD, there was a high CERS4, DEGS1, and SGMS2 expression in EAT and CERS3, CERS4 expression in PVAT. Among patients with CAD, expression of SPTLC1 in SAT and EAT, SPTLC2 in EAT, CERS2 in all studied AT, CERS4 and CERS5 in EAT, DEGS1 in SAT and EAT, ASAH1 in all studied AT, and SGMS1 in EAT were higher than in those with VHD. Protein levels of ceramide-metabolising enzymes were consistent with gene expression trends. The obtained results indicate activation of ceramide synthesis de novo and from sphingomyelin in cardiovascular disease, mainly in EAT, that contributes to the accumulation of ceramides in this location.
Overall, the manuscript is well written and very well structured, easy to read, with suggestive figures.
However, the authors can improve the Introduction and Discussion section, regarding the relationship between the epicardial and perivascular adipose tissue and atherogenesis. See the following articles:
1. https://doi.org/10.3390/diagnostics13010142
2. https://doi.org/10.3390/diagnostics13040624
3. https://doi.org/10.3390/diagnostics12112836
In conclusion, I want to congratulate the authors for their work.
Author Response

(The authors gave the same response as above.)

Round 2
Reviewer 1 Report
Dear Gruzdeva et al.,
Thank you for resubmitting your manuscript entitled “Expression of ceramide-metabolising enzymes in the heart adipose tissue of cardiovascular disease patients” to “International Journal of Molecular Sciences” following considering my comments and recommendations, which has tried to explain the possible mechanism of de-novo ceramide synthesis pathway in epicardial adipose tissue. The main strength of this study is comparing the expression level of various responsible genes and proteins in ceramide synthesis and utilization pathways by subcutaneous adipose tissue, epicardial adipose tissue, and perivascular adipose tissue. Overall, following revision of methods and results section, the manuscript is looking better.